# Unsupervised Analysis Reveals the Involvement of Key Immune Response Genes and the Matrisome in Resistance to BRAF and MEK Inhibitors in Melanoma

**DOI:** 10.3390/cancers16132313

**Published:** 2024-06-24

**Authors:** Feng Liu-Smith, Jianjian Lin

**Affiliations:** 1Department of Preventive Medicine, College of Medicine, University of Tennessee Health Science Center, Memphis, TN 38105, USA; jlin39@uthsc.edu; 2Department of Dermatology, College of Medicine, University of Tennessee Health Science Center, Memphis, TN 38105, USA

**Keywords:** melanoma, drug resistance, tumor microenvironment, matrisome, *PLXNC1*

## Abstract

**Simple Summary:**

Drug resistance is still an imminent issue for melanoma patients even after the use of BRAFi/MEKi combination therapy in clinic. Tumor heterogeneity and heterogeneity in treatment responses make it difficult to find consensus genes and pathways in resistance to therapy. This study used an objective method to analyze published gene expression data from pre-treated tumors and drug-resistant tumors, and identified possible targets and markers for resistant tumors, which centered on the *PLXNC1* gene, which promotes a pro-inflammatory tumor microenvironment.

**Abstract:**

Melanoma tumors exhibit a wide range of heterogeneity in genomics even with shared mutations in the MAPK pathway, including *BRAF* mutations. Consistently, adaptive drug resistance to *BRAF* inhibitors and/or *BRAF* plus MEK inhibitors also exhibits a wide range of heterogeneous responses, which poses an obstacle for discovering common genes and pathways that can be used in clinic for overcoming drug resistance. This study objectively analyzed two sets of previously published tumor genomics data comparing pre-treated melanoma tumors and BRAFi- and/or MEKi-resistant tumors. Heterogeneity in response to BRAFi and BRAFi/MEKi was evident because the pre-treated tumors and resistant tumors did not exhibit a tendency of clustering together. Differentially expressed gene (DEG) analysis revealed eight genes and two related enriched signature gene sets (matrisome and matrisome-associated signature gene sets) shared by both sets of data. The matrisome was closely related to the tumor microenvironment and immune response, and five out of the eight shared genes were also related to immune response. The *PLXNC1* gene links the shared gene set and the enriched signature gene sets as it presented in all analysis results. As the *PLXNC1* gene was up-regulated in the resistant tumors, we validated the up-regulation of this gene in a laboratory using vemurafenib-resistant cell lines. Given its role in promoting inflammation, this study suggests that resistant tumors exhibit an inflammatory tumor microenvironment. The involvement of the matrisome and the specific set of immune genes identified in this study may provide new opportunities for developing future therapeutic methods.

## 1. Introduction

Since the discovery of BRAF as a major driver mutation in melanoma [1], inhibitors of BRAF (BRAFis) have been developed and applied effectively in clinic settings [2,3,4]. Early results indicate that although BRAFis are generally effective and moderately extend life span in patients, drug resistance develops quickly. Hence, combination treatment of a BRAFi with a MEK inhibitor (MEKi) became a first-line choice of treatment, along with rapidly developed immune checkpoint inhibitors [5].

Combination treatment of a BRAFi and a MEKi still leads to the development of drug resistance, and tumors may re-grow after a period of regression [6]. There have been numerous studies revealing drug resistance mechanisms, such as oncogene amplification or mutation, alternative proliferation pathways and survival pathways, as well as innate and adaptive immune modulation pathways [7,8]. Genomic approaches have been used together with molecular validations in laboratories, and have identified various genomic components in resistance development [9,10]. For example, reactivation of MAPK pathways and rewiring through *NRAS* and mTORC1 pathways are found to play key roles in BRAFi resistance [11,12]. Discovering reactivation of the MAPK pathway led to the addition of MEKis in BRAFi treatment; however, BRAFi/MEKi combination treatment eventually still leads to adaptive drug resistance.

Heterogeneity of the melanoma genome and the alternative wiring of proliferation signaling upon drug treatment complicate the situation. This study aims to describe such complexity using unsupervised computational re-analysis on published tumor genomics data, and discovers that a unique set of shared genes in resistance tumors. Furthermore, the matrisome and its associated gene signatures are found to be the only significant enriched gene set among all MSigDB gene sets, which was not previously reported.

The term “matrisome” was used to collectively describe extracellular matrix (ECM) components, which comprise ~300 proteins in mammalian cells [13]. Matrisome-associated components include ECM-binding factors, ECM-modifying enzymes, and other ECM-associated proteins. Hence, matrisome and matrisome-associated pathways play crucial roles in cell proliferation, tumor microenvironment modification, tumor immune response, cell differentiation, cell migration, and metastasis [14,15]. Recently, a panel of six matrisome-related genes were found to be associated with prognostic outcomes in SCCHN [16], and the matrisome can become an effective treatment target [17].

## 2. Material and Methods

Genomic datasets (GSE50509 and GSE61992) were downloaded from Gene Expression Omnibus (GEO) at the NCBI website (https://www.ncbi.nlm.nih.gov/geo/, accessed on 10 February 2024) using R package GEOquery() (Version 2.72.0) [18]. Both expression datasets were generated by the GPL10558 platform, Illumina HumanHT-12 V4.0 expression beadchip [9,10,11]. While most phenotype data was downloaded from the GEO website, some patient information was extracted from original publications [9,10,11]. Principal component analysis (PCA) was performed using the R-built-in *prcomp*() function while pheatmap was used to generate a heat map of clusters [19]. Differential expression analysis was performed using the limma package (Version 3.60.2) [20], and GSEA analysis was performed using the fgsea package (Version 1.30.0) in R [21].

The TCGA-SKCM dataset was downloaded from the Broad Institute firehouse website: (https://gdac.broadinstitute.org/, accessed on 10 February 2024). All data processing was carried out in R.

Data processing: The downloaded data was adjusted such to ensure all expression values were positive, and then the values were log2 transformed. For limma analysis, genes were ranked by a formula: “sign(logFC)*(−log10(*p*.Value))”. Significant genes were retained by a nominal *p* value of <0.05 and a 1.5-fold change in either direction. A Venn diagram was drawn by Gene Venn program (https://www.bioinformatics.org/gvenn/, accessed on 10 February 2024).

Vemurafenib-resistant cells, RNA isolation, and qRT-PCR: The establishment of vemurafenib-resistant cell lines was described before [22]. Briefly, the parental melanoma cell lines SK-Mel28 and 1205Lu were cultured in increasing concentrations of vemurafenib for 4–8 weeks. In the end, the cells were able to proliferate normally in a vemurafenib concentration equal to or greater than the IC50s of the parental cell lines. RNA was isolated using the PureLink™ RNA Mini Kit (Invitrogen, Carlsbad, CA, USA, cat #12183018A), and then treated with DNase before being subjected to one-step qRT-PCR. The GAPDH gene was used as an internal control (primers: 5′-CCA CTCCTC CAC CTT TGA CGC -3′; 5′-GAC TGAGTG TGG CAG GGACTC-3′). The primers for PLXNC1 were designed by NCBI Primer3 (forward primer: 5′-GGT GGC AAT TCA TTC TGT GCT T-3′, reverse primer: 5′-TGA GGA TGA CAA TGG AGG CAA A-3′). Relative mRNA levels were calculated by the DDCt method, as described before [23]. The experiments were repeated twice, with three replicates each time. The two-sided *p* values were calculated by a Student’s *t*-test based on whether the DDCt value was different than 0 (Stata/SE18.0, College Station, TX, USA).

## 3. Results

### 3.1. The Clustering of Tumors and Patients Reveals No Clustering by Pre-Treatment or Post-Treatment, or by Any Other Criteria

For an initial examination of data, expression dataset GSE50509 was used for evaluation. This dataset constitutes 61 tumors from 21 patients, among which 28 were from pre-BRAFi treatment while 33 were progressed drug-resistant tumors (Table 1). Some patients had more than one tumor before treatment and/or after treatment (Appendix A).

First, the entire dataset was used for analysis, with an assumption that gene expression levels below the median of the normalized beadchip readings were not expressed and hence removed. Unexpectedly, the heatmap generated by the pheatmap package showed no clusters based on any of the categorical variables (Figure 1a), including pre-treated tumors vs. resistant tumors, age, sex, and response to treatment. To validate this result, PCA analysis was used to show the relative relationship of tumors and patients (Figure 1b). Similar to the heatmap results, the pre-treated tumors were mixed with progressed tumors, exhibiting no tendency of grouping by treatment. For example, one of the two progressed tumors in Patient 10 (treated with dabrafenib, Appendix A) is quite close to the original two tumors from the same patient, while the other one showed a distant location. Patient 11 (treated with dabrafenib) has six tumors; two progressed tumors are closer to two of the pre-treated ones, but the other two progressed ones are distant.

In order to validate this result, a second dataset GSE61992 was used for a similar analysis. This dataset contained the gene expression profiles of 26 tumors from 11 patients who were treated with combined dabrafenib and trametinib (Table 1 and Appendix A). Again, tumors were not grouped by treatment, but they seemed to be grouped by patient in this dataset, regardless of being treated or not (Figure 1c,d). The tumors were not grouped by treatment or type of BRAF mutation.

### 3.2. Analysis Using the Melanoma-Specific Gene Set also Returned No Significant Clustering

Recognizing that the presumption above, which assumes that 50% of genes are not expressed in the dataset, is overly arbitrary, the TCGA-SKCM data was then utilized to identify all genes specifically expressed in melanoma tumors. All genes with an RSEM (RNA-Seq by Expectation-Maximization) expression level greater than one across all tumors in the TCGA-SKCM dataset were considered to be expressed in melanoma tumors. A total of 17,895 genes remained, which were termed melanoma-specific genes. Next the melanoma-specific gene sets from the GSE50509 and GSE61992 datasets were extracted, and the heatmap and PC analysis were used for clustering again. The results were identical to the first analysis, with no apparent clustering of pre-treated tumors and resistant tumors.

### 3.3. Unsupervised Differential Gene Expression Analysis

Because of the above heterogeneity of gene expression profiles, differential gene expressions (DEGs) between pre-treated tumors and progressed tumors were obtained by nominal *p* values with a cutoff of 1.5-fold changes. Multiple comparison adjustment was not used as this would return no significant DEGs, either using 50% of all genes or using the melanoma-specific genes.

Limma analysis resulted in 380 significant genes (unadjusted *p* < 0.05) in the DEG set for GSE50509 and 569 significant genes (unadjusted *p* < 0.05) for GSE61992 (Appendix A, Appendix A). A Venn diagram was used to examine the shared genes in these two sets (Appendix A). There were only 12 genes shared by these two datasets (Table 2), among which 8 genes had changes in the same direction in the two datasets: *ARHGAP18* (Rho GTPase-activating protein 18), *FOLR2* (Floate Receptor beta), *IFI44* (Interferon-Induced Protein 44), *LEF1* (Lymphoid Enhancer Binding Factor 1), *OLR1* (Oxidized Low-Density Lipoprotein Receptor 1), *PKD2* (Polycystin 2, Transient Receptor Potential Cation Channel), *PLXNC1* (Plexin C1), and *SAMD9* (Sterile Alpha Motif Domain-Containing 9).

### 3.4. Gene Set Enrichment Analysis

Each DEG set was ranked and then used for gene set enrichment analysis (GSEA). Each molecular signature set collected in MSigDB was separately used for GSEA analysis, including Hallmark gene signatures and C1 to C8 gene signatures [24,25,26]. Results for GSE50509 and GSE61992 are listed in Table 3. With the adjusted *p*-value of 0.05 as a cutoff point, there are 26 and 22 significantly enriched gene sets for the GSE50509 and GSE61992 datasets, respectively. Interestingly only two related gene sets are found in both datasets: NABA_Matrisome and NABA_Matrisome_Associated. However, the enrichment is in two directions: the normalized enrichment scores in the GSE50509 were −2.38 and −1.83 for NABA_Matrisome and NABA_Matrisome_Associated, respectively, while that for GSE61992 were 2.35 and 2.12, respectively (Table 3, in bold). When the leading-edge components are examined, the two gene sets actually have different leading-edge genes except for one shared gene, *PLXNC1*, which shows the same regulation direction in two datasets. *PLXNC1* was up-regulated in the resistant tumors by 1.73- and 1.51-fold in the GSE50509 and GSE61992 sets, respectively. Therefore, this shared matrisome enrichment is at least not contradictory to each other; rather, they are consistent.

### 3.5. Validation of PLXNC1 Over-Expression in BRAFi-Resistant Cells

To validate that *PLXNC1* is indeed up-regulated in BRAFi-resistant cells, we performed a qRT-PCR experiment using two pairs of cell lines: the parental SK-Mel28 and 1205Lu cells, as well as their corresponding vemurafenib-resistant cell lines. These resistant cell lines were established in our laboratory before and reported elsewhere [22]. The qRT-PCR results are shown in Figure 2. In comparison to the parental cell lines, SK-Mel28-R (SK-Mel28 resistant cell line) showed about a 4.1-fold ± 0.5 higher expression of *PLXNC1* (*p* = 0.04) while 1205Lu-R (1205Lu resistant cells) showed a 6.2-fold ± 0.7 higher expression of *PLXNC1* compared to 1205Lu cells (*p* = 0.02, Figure 2).

## 4. Discussion

It is desirable to discover consensus gene regulation in tumors resistant to BRAFi and MEKi treatment for the sake of developing new treatment methods. For example, reactivation of the MAPK pathway was discovered following BRAFi treatment [9], which resulted in the combination treatment of a BRAFi and a MEKi. While there are many other similar attempts, the actual results varied with limited clinical advancement, perhaps due to the extreme heterogeneity of tumors and drug resistance responses. Such heterogeneity was reported before, but the main focus was on the MAPK pathway and melanoma-related oncogenic pathways and tumor suppressors, such as *PI3K*, *PTEN*, *AKT*, and *CDKN2A* [27], or amplification of *BRAF* and mutations of the *RAS* family [12,28]. This study used published datasets and analyzed gene expression objectively and discovered that while there were a few immune-related genes showing alteration in resistant tumors, the matrisome seemed to be a significant player.

First, the heterogeneity of gene expression profiling was demonstrated by PCA analysis and a heatmap, in which the resistant tumors were mingled in with the early dissected tumors or untreated tumors, without a distinct clustering pattern. For this reason, nominal *p* values had to be used for DEG analysis. Only 12 genes were shared in the two datasets, among which 8 genes were in the same direction.

Five genes (*IFI44*, *FLOR2*, *OLR1*, *PLXNC1*, and *SAMD9*) among these eight genes were related to tumor-associated macrophages or tumor immune response. *IFI44* is an interferon-a inducible protein which suppresses the immune response during viral infection [29,30]. This may be consistent with a previous report that BRAFi impacted the tumor microenvironment where the number of immune cells were reduced during BRAFi treatment [31]. The *FLOR2* gene is a marker for tumor-associated macrophages and is identified as an anti-inflammatory gene [32]. *FLOR2* is down-regulated in resistant tumors, suggesting an increased inflammatory tumor microenvironment, which is consistent with the effect of increased *IFI44* gene expression in resistant tumors. As for the *OLR1* gene, a recent scRNA-seq analysis in head and neck tumors showed that OLR1 was specifically expressed on tumor-associated macrophages, and was significantly associated with worse overall survival of patients [33]. Similarly, *PLXNC1*, a member of the Plexin family composed of transmembrane domains, was also shown to be significantly associated with M2 macrophages, and indicated poor outcome in stomach adenocarcinoma and acute myeloid leukemia [34,35]. The direction of gene regulation of these four genes points to poor overall survival, which is consistent with resistance to BRAFi or BRAFi/MEKi treatment.

The *ARHGAP18* gene is involved in endothelial cell regulation and tumor angiogenesis [36], while the PKD2 protein which contains six transmembrane domains, plays a key role in autophagy, which is an intracellular degradation process under stressful stimuli [37]. The *LEF1* transcriptional factor is multi-functional, and its involvement in BRAFi/MEKi resistance has not been well established [38], but it may be related to epithelial–mesenchymal transition.

The only two shared enrichment gene sets are matrisome and matrisome-associated genes in the two GSE datasets. The matrisome is a set of gene signatures established by a proteosome method to identify extracellular matrix components, i.e., an ECM (extracellular matrix) proteome [15]. Matrisome-associated genes are proteins regulating or modulating the ECM proteome [15]. Together there are ~1000 matrisome and matriosome-associated genes. Both GSE datasets identified matrisome and matrisome-associated genes in the GSEA analysis, with one single shared gene in the leading gene set, *PLXNC1*, which was in the same direction of regulation. As the ECM or matrisome, indeed, defines the tumor microenvironment, which is closely related to immune regulation, inflammation, tumor invasiveness, and metastasis [14,16], it is no surprise that this study found that the matrisome exhibited a significant change in drug-resistant tumors.

Since *PLXNC1* appeared in the shared gene set as well as in the shared enrichment gene set, its function in cellular signaling was further examined. *PLXNC1* is a receptor for the *SEMA7A* protein, which regulates a wide range of tumor cell functions, including proliferation, invasion, migration, and angiogenesis. *PLXNC1* is induced during acute inflammation and is expressed in the nervous system. Knockout of *PLXNC1* led to decreased inflammatory responses and improved survival in mice [39,40]. We showed that *PLXNC1* was indeed up-regulated in vemurafenib-resistant SK-Mel28 and 1205Lu melanoma cells.

*PLXNC1* expression was lost in metastatic melanoma as compared to matched primary tumors [41], and, therefore, *PLXNC1* in melanoma is considered to be a tumor suppressor. A further piece of supporting evidence is found in a mouse model of melanoma, where *PlxnC1* expression was positively correlated with *Ednrb*, suggesting a role in the *Ednrb/EDNRB*-mediated suppressive effect on melanoma development [42]. However, in hepatocellular carcinoma and stomach cancer, *PLXNC1* promotes cancer cell proliferation and metastasis, and is associated with overall poorer survival [43,44]. In stomach cancer, *PLXNC1* was significantly associated with M2 macrophages and poor outcome [34]. In the context of the BRAFi and MEKi treatment condition, increased expression of *PLXNC1* perhaps indicated an increased inflammatory tumor microenvironment.

It was reported that BRAFi can create a pro-inflammatory tumor microenvironment via reduced recruitment of immunosuppressive cells and increased immune effector cells, as well as an increased ability of immune effector cells to trigger cancer killing [45]. This inflammatory tumor microenvironment could be subjective to dynamic regulation when drug resistance develops [45].

As the overall sample size is not large enough in this study, the resulting common genes and signature gene sets need further validation with larger datasets, or validation in a laboratory. If results are validated, it is possible to develop *PLXNC1* as a progression biomarker for resistant tumors, or as a future treatment target.

## 5. Conclusions

Consistent with the functions of the eight shared genes found in the two datasets, the matrisome changes with tumor progression and is also connected to the tumor immune response [14,16], and is a possible target for developing targeted therapy. Therefore, overall, this study reveals previously unknown aspects of the BRAFi and MEKi resistance phenotype, i.e., modulation of the matrisome and a few specific immune-related genes that were not identified in previous studies.

## Figures and Tables

**Figure 1 cancers-16-02313-f001:**
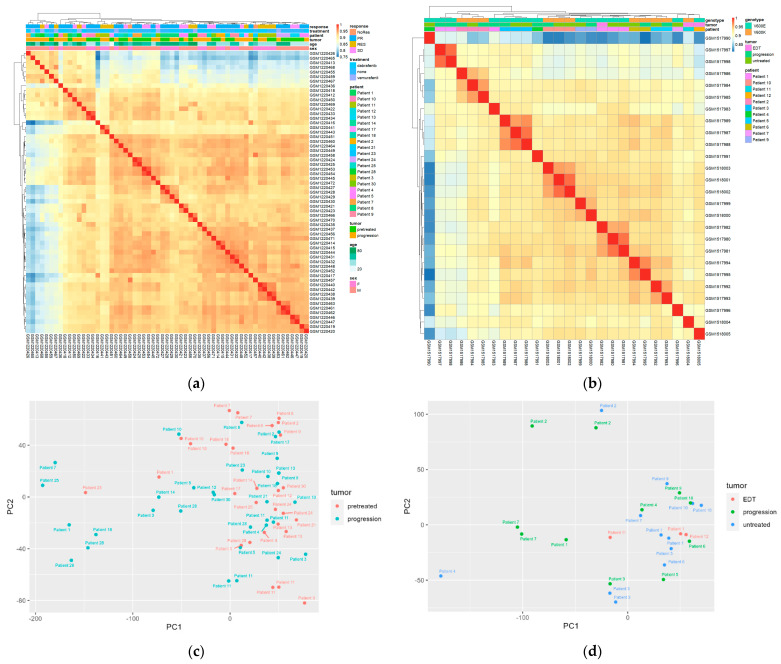
Melanoma tumors and patients are not grouped by drug resistance. (**a**,**b**) Heatmap of gene expression in GSE50509 and GSE61992, respectively. In the “response” variable, NoRes, no response; PR, partial response; RES, response; SD, stable disease. Tumors are colored by “pretreatment” and “progression” (resistant tumors). (**c**,**d**) Principal component analysis of the two datasets based on patients and treatment. EDT: early during treatment when the tumors were excised. Some patients have multiple tumors, either from pre-treatment or progression.

**Figure 2 cancers-16-02313-f002:**
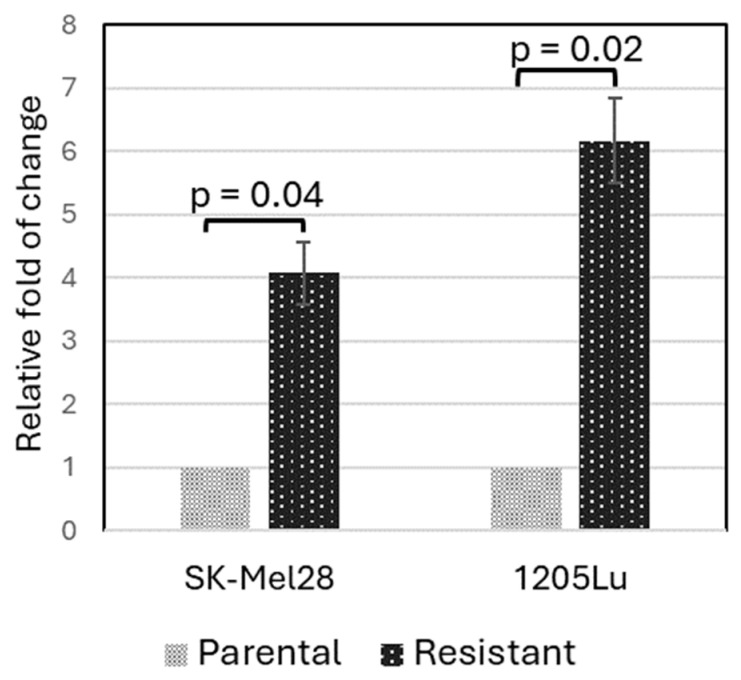
PLXNC1 is up-regulated at an mRNA level in BRAFi-resistant melanoma cell lines. The qRT-PCR experiment was performed with GAPDH as an internal control, and the DDCt method was used to calculate the relative expression levels of the resistant cells normalized to the parental control cells.

**Table 1 cancers-16-02313-t001:** Tumor characteristics from the two datasets.

Dataset	None or EDT *	Vemurafenib	Dabrafenib	All
GSE50509	28	8	25	61
GSE61992	14	0	10 **	24
Total	42	8	35	85

* EDT: early during treatment; ** indicates Dabrafenib/Trametinib combination treatment.

**Table 2 cancers-16-02313-t002:** The 12 shared genes in the two datasets.

	GSE50509			GSE61992				
SYMBOL	logFC	AveExpr	*p*. Value	logFC	AveExpr	*p.* Value	Band	Gene Description
*ARHGAP18*	0.61	6.47	0.027	0.70	5.36	0.025	6q22.33e	Rho GTPase-activating protein 18
*FOLR2*	−0.69	7.32	0.012	−0.64	5.71	0.017	11q13.4a	folate receptor beta
*IFI44*	0.77	8.22	0.039	1.36	5.98	0.020	1p31.1e	Interferon-induced protein 44
*LEF1*	0.69	9.78	0.005	0.68	8.62	0.039	4q25b	lymphoid enhancer binding factor 1
*OLR1*	0.79	6.56	0.011	0.81	6.34	0.013	12p13.2c	oxidized low-density lipoprotein receptor 1
*PKD2*	0.80	7.96	0.004	0.75	4.95	0.029	4q22.1b	polycystin 2, transient receptor potential cation channel
*PLXNC1*	0.79	6.37	0.023	0.60	3.96	0.034	12q22c	Plexin C1
*SAMD9*	0.84	6.91	0.017	1.16	4.95	0.031	7q21.2b	sterile alpha motif domain-containing 9
*SLAMF9*	−0.82	5.88	0.027	0.62	9.82	0.039	1q23.2c	SLAM family member 9
*TP53TG1*	−0.77	6.69	0.001	0.65	5.45	0.048	7q21.12	TP53 target 1
*CEBPA*	−0.72	7.63	0.026	0.93	8.39	0.011	19q13.11b	CCAAT enhancer-binding protein alpha
*CKB*	−0.87	7.06	0.020	0.93	4.39	0.003	14q32.33a	creatine kinase B

**Table 3 cancers-16-02313-t003:** The GSEA results. Bolded: shared enriched pathways by the two datasets.

Dataset	Signature	Pathway	*p* val	padj	ES	NES	Size	Leading Edge Genes
GSE50509	C1	chr19q13	0.0000	0.0000	−0.820	−3.386	11	*TMEM145*, *PAFAH1B3*, *BLVRB*, *LIN7B*, *CNFN*, *LILRA2*, *LRFN3*, *CEBPA*, *ZNF296*, *NCCRP1*
	**C2:CP**	**NABA_MATRISOME**	**0.0010**	**0.0021**	**−0.714**	**−2.380**	**15**	** *F12*, *CCL18*, *POSTN*, *SFRP4*, *SCUBE2*, *CXCL8*, *PDGFD*, *SULF1*, *PRG4*, *EMCN*, *PLXNC1*, *SPP1*, *SPARCL1*, *ANXA1 PLXNC1*, *SPP1*, *SPARCL1*, *ANXA1* **
	**C2:CP**	**NABA_MATRISOME_ASSOCIATED**	**0.0142**	**0.0142**	**−0.563**	**−1.829**	**11**	** *F12*, *CCL18*, *SFRP4*, *SCUBE2*, *CXCL8*, *PDGFD*, *SULF1*, *EMCN*, *PLXNC1* **
	C3:“MIR:MIRDB”	MIR106B_5P	0.0002	0.0336	0.451	2.338	31	*KLF9*, *SOS1*, *BBX*, *IL1RAP*, *SSH1*, *KAT2B*, *PKD2*, *JAK1*, *NPAT*, *REV3L*, *ANO6*, *TMX3*, *PAFAH1B1*, *RBM12B*, *PDE3B*, *BMPR2*, *CALD1*, *SPTY2D1*, *ARID4B*, *PLEKHA3*
	C3:“MIR:MIRDB”	MIR20A_5P	0.0002	0.0336	0.451	2.338	31	*KLF9*, *SOS1*, *BBX*, *IL1RAP*, *SSH1*, *KAT2B*, *PKD2*, *JAK1*, *NPAT*, *REV3L*, *ANO6*, *TMX3*, *PAFAH1B1*, *RBM12B*, *PDE3B*, *BMPR2*, *CALD1*, *SPTY2D1*, *ARID4B*, *PLEKHA3*
	C3:“MIR:MIRDB”	MIR106A_5P	0.0003	0.0484	0.457	2.327	29	*KLF9*, *SOS1*, *BBX*, *IL1RAP*, *SSH1*, *KAT2B*, *PKD2*, *JAK1*, *NPAT*, *REV3L*, *ANO6*, *TMX3*, *RBM12B*, *PDE3B*, *BMPR2*, *CALD1*, *SPTY2D1*, *ARID4B*, *PLEKHA3*
	C3:“TFT:TFT_Legacy”	TAATTA_CHX10_01	0.0006	0.0493	0.423	2.144	31	*TSPAN7*, *SORBS2*, *TRIM24*, *CDH19*, *PRRX1*, *PPFIBP1*, *AMD1*, *MEF2C*, *OSBPL8*, *SECISBP2L*, *IFI16*, *BMPR2*, *CALD1*, *BAZ1A*, *CAB39*, *ZC3H7A*, *STXBP3*, *MYO1B*, *CPEB4*, *PELI2*, *EPB41L3*
	C5:“GO:BP”	GOBP_CIRCULATORY_SYSTEM_DEVELOPMENT	0.0000	0.0145	0.470	2.484	35	*DUSP6*, *SOS1*, *SORBS2*, *PRRX1*, *MATR3*, *ARID2*, *EP300*, *KAT2B*, *PKD2*, *MEF2C*, *JAK1*, *SPRY2*, *LEF1*, *ROCK1*, *HTR2B*, *PDE3B*, *BMPR2*, *C1GALT1*, *ITGAV*, *CALD1*
	C5:“GO:BP”	GOBP_HEART_DEVELOPMENT	0.0001	0.0145	0.573	2.445	18	*DUSP6*, *SOS1*, *SORBS2*, *MATR3*, *ARID2*, *EP300*, *KAT2B*, *PKD2*, *MEF2C*, *ROCK1*, *HTR2B*, *BMPR2*
	C5:“GO:BP”	GOBP_DEVELOPMENTAL_GROWTH	0.0002	0.0145	0.567	2.418	18	*DUSP6*, *SOS1*, *SORBS2*, *IQGAP1*, *ARID5B*, *ARID2*, *EP300*, *MEF2C*, *SPRY2*, *MTM1*, *PAFAH1B1*, *BMPR2*
	C5:“GO:BP”	GOBP_ANIMAL_ORGAN_MORPHOGENESIS	0.0001	0.0145	0.511	2.404	24	*CSGALNACT1*, *SOS1*, *SNX10*, *PRRX1*, *ARID5B*, *ARID2*, *EP300*, *PKD2*, *MEF2C*, *SPRY2*, *LEF1*, *HTR2B*, *PAFAH1B1*, *BMPR2*
	C5:“GO:BP”	GOBP_REGULATION_OF_PHOSPHORUS_METABOLIC_PROCESS	0.0002	0.0145	0.448	2.324	33	*DUSP6*, *SOS1*, *IQGAP1*, *ACSL3*, *EP300*, *KAT2B*, *PKD2*, *MEF2C*, *SPRY2*, *TNIK*, *PTPN13*, *OSBPL8*, *GNAQ*, *ROCK1*, *HTR2B*, *BMPR2*, *BST1*, *CAB39*, *CD44*, *SASH1*
	C5:“GO:BP”	GOBP_VASCULATURE_DEVELOPMENT	0.0004	0.0266	0.473	2.254	25	*SOS1*, *PRRX1*, *ARID2*, *PKD2*, *MEF2C*, *JAK1*, *SPRY2*, *LEF1*, *ROCK1*, *PDE3B*, *BMPR2*, *C1GALT1*, *ITGAV*, *CALD1*, *SASH1*, *HECTD1*, *ANXA1*
	C5:“GO:BP”	GOBP_TISSUE_DEVELOPMENT	0.0005	0.0281	0.358	2.211	55	*MCOLN3*, *CSGALNACT1*, *SOS1*, *SORBS2*, *IQGAP1*, *SNX10*, *PRRX1*, *ARID5B*, *ARID2*, *EP300*, *PKD2*, *MEF2C*, *GSTM3*, *SPRY2*, *LEF1*, *UGCG*, *MTM1*, *ROCK1*, *ANO6*, *HTR2B*, *PAFAH1B1*, *BMPR2*, *C1GALT1*, *ITGAV*
	C5:“GO:BP”	GOBP_DEPHOSPHORYLATION	0.0008	0.0369	0.578	2.196	13	*NT5C3A*, *DUSP6*, *IQGAP1*, *SSH1*, *MEF2C*, *PTPN13*, *MTM1*, *ROCK1*
	C5:“GO:BP”	GOBP_POSITIVE_REGULATION_OF_PHOSPHORUS_METABOLIC_PROCESS	0.0010	0.0404	0.496	2.126	19	*IQGAP1*, *ACSL3*, *PKD2*, *MEF2C*, *SPRY2*, *TNIK*, *OSBPL8*, *ROCK1*, *HTR2B*, *BMPR2*, *CAB39*, *CD44*, *SASH1*
	C5:“GO:BP”	GOBP_POSITIVE_REGULATION_OF_TRANSFERASE_ACTIVITY	0.0011	0.0404	0.574	2.116	12	*IQGAP1*, *PKD2*, *SPRY2*, *OSBPL8*, *HTR2B*, *BMPR2*, *CAB39*, *SASH1*, *HNRNPA2B1*, *ARRDC4*, *TOM1L1*
	C5:“GO:BP”	GOBP_TUBE_MORPHOGENESIS	0.0013	0.0456	0.451	2.183	26	*SOS1*, *PRRX1*, *ARID2*, *PKD2*, *MEF2C*, *JAK1*, *SPRY2*, *LEF1*, *ROCK1*, *PDE3B*, *BMPR2*, *C1GALT1*, *ITGAV*, *CALD1*, *SASH1*, *HECTD1*, *ANXA1*
	C5:“GO:BP”	GOBP_EMBRYO_DEVELOPMENT	0.0016	0.0490	0.409	2.088	30	*SOS1*, *TBC1D23*, *PRRX1*, *ARID2*, *EP300*, *PKD2*, *MEF2C*, *SPRY2*, *LEF1*, *NPAT*, *ROCK1*, *HTR2B*, *PAFAH1B1*, *BMPR2*, *ITGAV*
	C5:HPO	HP_DOWNSLANTED_PALPEBRAL_FISSURES	0.0002	0.0343	0.706	2.192	10	*TSPAN7*, *CSGALNACT1*, *MPDZ*, *SOS1*, *PRRX1*, *ARID2*, *EP300*
	C8	FAN_OVARY_CL1_GPRC5A_TNFRS12A_HIGH_SELECTABLE_FOLLICLE_STROMAL_CELL	0.0003	0.0338	0.553	2.183	15	*ARID5B*, *AMD1*, *GSTM3*, *IFI16*, *UBL3*, *PLOD2*, *CALD1*, *SGCE*, *HNRNPA2B1*, *CTNNAL1*, *RHOBTB3*, *DDX21*, *EIF3J*
	C8	MANNO_MIDBRAIN_NEUROTYPES_HPERIC	0.0013	0.0342	0.418	2.157	31	*DUSP6*, *SOS1*, *BBX*, *IQGAP1*, *MATR3*, *ARID2*, *SSH1*, *EP300*, *SPATA13*, *KAT2B*, *JAK1*, *SH3KBP1*, *ST3GAL5*, *NPAT*, *VPS26A*, *SNRK*, *OSBPL8*, *REV3L*, *BTBD1*, *ROCK1*, *TYW3*, *YTHDC2*, *IFI16*, *PAFAH1B1*, *ANKRD49*, *RAD21*, *SLC2A3*, *TBC1D4*, *BAZ1A*, *CAB39*, *TRAPPC8*, *CD44*, *ARID4B*, *HNRNPA2B1*, *ITPR1*, *ANXA1*, *TMEM123*, *BCLAF1*, *U2SURP*, *MYCBP2*, *SAMD9*, *PDS5A*, *FNBP4*, *ACAP2*, *ACTR2*
	C8	BUSSLINGER_GASTRIC_IMMUNE_CELLS	0.0009	0.0342	0.320	2.092	73	*DUSP6*, *SOS1*, *BBX*, *IQGAP1*, *MATR3*, *ARID2*, *SSH1*, *EP300*, *SPATA13*, *KAT2B*, *JAK1*, *SH3KBP1*, *ST3GAL5*, *NPAT*, *VPS26A*, *SNRK*, *OSBPL8*, *REV3L*, *BTBD1*, *ROCK1”*
	C8	FAN_EMBRYONIC_CTX_BRAIN_ENDOTHELIAL_2	0.0017	0.0342	0.531	2.045	14	*GNG12*, *JAK1*, *ST3GAL5*, *SNRK*, *CALD1*, *SASH1*, *SGCE*, *MYO1B*, *ITPR1*, *FRMD6*, *PHACTR2*, *SPARCL1*
	C8	RUBENSTEIN_SKELETAL_MUSCLE_SMOOTH_MUSCLE_CELLS	0.0016	0.0342	0.514	2.030	15	*BBX*, *SORBS2*, *ARID5B*, *MEF2C*, *TCEA1*, *TAX1BP1*, *SNRK*, *ROCK1*, *CALD1*, *SGCE*, *SPARCL1*, *NSA2*, *PCNP*, *PRSS23*
	C8	MURARO_PANCREAS_MESENCHYMAL_STROMAL_CELL	0.0021	0.0362	0.402	2.072	31	*DUSP6*, *KLF9*, *PRRX1*, *PPFIBP1*, *ARID5B*, *FAM114A1*, *SSH1*, *JAK1*, *SH3KBP1*, *ANO6*, *IFI16*, *PLOD2*, *ITGAV*, *SLC2A3*, *CALD1*, *BAZ1A*, *SASH1*, *ANXA1*, *FRMD6*, *DDX21*
GSE61992	C2:CGP	CHICAS_RB1_TARGETS_SENESCENT	0.0000	0.0127	0.425	2.743	34	*PITPNC1*, *IFI6*, *HIPK2*, *TREM1*, *APCDD1L*, *VEGFA*, *TXNIP*, *IL6*, *CA12*, *GBP3*, *GK*, *FEZ1*, *CTSS*, *ODC1*, *CXCL5*, *PSMB8*, *EIF4EBP1*, *HLA-B*, *PRR11*, *CXCL10*, *IFITM1*
	C2:CGP	PEREZ_TP53_AND_TP63_TARGETS	0.0002	0.0406	0.569	2.459	14	*MYLIP*, *NEFL*, *CKB*, *GPT2*, *SEMA4D*, *RBP7*, *PPP1R16B*, *ADAP2*, *MAFB*, *RASGRP1*
	C2:CGP	YOSHIMURA_MAPK8_TARGETS_UP	0.0002	0.0406	0.298	2.329	52	*NEFL*, *RGS22*, *ENPP6*, *CD247*, *ZNF226*, *LGALS9*, *FOXM1*, *PTPN3*, *KCNK1*, *CSNK1D*, *NELL2*, *FEZ1*, *ITGB3*, *CD74*, *CTSS*, *INPP5D*, *MAFB*, *PSMB8*, *TAP2*, *PLEK*, *DUSP4*, *HLA-DQA1*, *LYZ*, *KLRB1*, *CCL5*, *RGS18*, *GZMB*, *NMU*, *PFKFB3*, *CD52*
	C2:CGP	BLANCO_MELO_COVID19_SARS_COV_2_POS_PATIENT_LUNG_TISSUE_UP	0.0003	0.0471	0.411	2.318	26	*OAS2*, *IFI6*, *TREM1*, *IFIT1*, *OLR1*, *TNF*, *OAS3*, *WAS*, *TYROBP*, *CD37*, *HSH2D*, *CXCL10*, *PLEK*, *IFITM1*, *SAMD9*, *PLAC8*, *RGS18*, *RTP4*
	**C2:CP**	**NABA_MATRISOME_ASSOCIATED**	**0.0003**	**0.0015**	**0.413**	**2.348**	**43**	** *SCUBE3*, *PLOD1*, *SEMA4D*, *VEGFA*, *IL6*, *LGALS9*, *ADAMTSL2*, *TNF*, *CTSS*, *CXCL5*, *PLXDC2*, *CTSO*, *CXCL10*, *CTSW*, *P4HA2*, *ADAMTS9*, *ANGPTL7*, *SERPINI1*, *PLXNA2*, *GPC3*, *FLT3LG*, *CCL5* **
	**C2:CP**	**NABA_MATRISOME**	**0.0010**	**0.0025**	**0.374**	**2.120**	**50**	** *SCUBE3*, *PLOD1*, *SEMA4D*, *VEGFA*, *IL6*, *LGALS9*, *ADAMTSL2*, *NELL2*, *TNF*, *CTSS*, *CXCL5*, *PLXDC2*, *CTSO*, *CXCL10*, *CTSW*, *P4HA2*, *ADAMTS9*, *ANGPTL7*, *SERPINI1*, *PLXNA2*, *GPC3*, *FLT3LG*, *CCL5*, *PLXNC1*, *HAPLN1* **
	C3:“MIR:MIRDB”	MIR548P	0.0001	0.0142	0.611	2.678	14	*DSP*, *RRAS2*, *PITPNC1*, *VEGFA*, *CEBPA*, *CSNK1D*, *IPPK*, *CXCL5*, *PLXDC2*, *MAFB*
	C3:“MIR:MIR_Legacy”	MIR548P	0.0000	0.0103	0.611	2.677	14	*DSP*, *RRAS2*, *PITPNC1*, *VEGFA*, *CEBPA*, *CSNK1D*, *IPPK*, *CXCL5*, *PLXDC2*, *MAFB*
	C3:“TFT:GTRD”	ZNF513_TARGET_GENES	0.0002	0.0311	0.493	2.500	19	*SCUBE3*, *CKB*, *PLOD1*, *SEMA4D*, *JPH1*, *TXNIP*, *CD6*, *NPAS1*, *HKDC1*, *HLA-F*, *ERC2*, *BOC*, *ADAMTS9*, *PKD2*
	C5:“GO:BP”	GOBP_HOMEOSTATIC_PROCESS	8.00 × 10^−6^	0.0056	0.301	2.545	73	*MYLIP*, *TRIM6*, *SLC24A3*, *CKB*, *CSF1R*, *RPE65*, *IFI6*, *HIPK2*, *VEGFA*, *JPH1*, *IFIT1*, *IL6*, *CEBPA*, *LGALS9*, *PTPN3*, *CA12*, *NELL2*, *HCLS1*, *TNF*, *ITGB3*, *GPR65*, *CD74*, *INPP5D*, *HKDC1*, *NOD2*, *MAFB*, *P2RY8*, *STAT1*, *NTSR1*, *HSH2D*, *CXCL10*, *CHRNA1*, *CD55*, *METRNL*, *PKD2*, *LYZ*, *SLC40A1*, *LDLR*, *PLAC8*, *CCL5*, *HK2*, *NMU*
	C5:“GO:CC”	GOCC_INTRINSIC_COMPONENT_OF_PLASMA_MEMBRANE	0.0006	0.0502	0.239	2.214	83	*SLC6A15*, *SLC24A3*, *KIR2DL3*, *TSPAN13*, *KCNF1*, *IL2RB*, *CSF1R*, *SEMA4D*, *IL6*, *ADORA2B*, *TSPAN32*, *KCNK1*, *IL18RAP*, *OLR1*, *CD4*, *TNF*, *TLR1*, *ITGB3*, *GPR65*, *MPZL1*, *CD6*, *ITGAL*, *NOD2*, *PCDHB13*, *P2RY8*, *HLA-B*, *NTSR1*, *TYROBP*, *CD37*, *KLRD1*, *CHRNA1*, *GPNMB*, *BOC*, *PKD2*, *HLA-DQA1*, *SLC7A7*, *PLXNA2*, *GPC3*, *SLC40A1*, *LDLR*, *LRRC8C*, *PLXNC1*
	C6	STK33_NOMO_UP	0.0008	0.0528	0.425	2.140	21	*SLC24A3*, *FCHO2*, *RNASEL*, *SEMA4D*, *TREM1*, *HAVCR2*, *PTPN3*, *OLR1*, *GBP3*, *TLR1*, *EOMES*
	C7:VAX	NAKAYA_PBMC_FLUARIX_FLUVIRIN_AGE_18_50YO_7DY_DN	0.0002	0.0149	0.573	2.473	14	*VEGFA*, *HAVCR2*, *SIK1*, *CD6*, *HLA-F*, *MAFB*, *NR4A2*, *CD55*, *METRNL*, *ITPRIP*, *SAMSN1*, *PFKFB3*
	C8	TRAVAGLINI_LUNG_EREG_DENDRITIC_CELL	7.99 × 10^−5^	0.0155	0.334	2.513	48	*HLA-DOA*, *CSF1R*, *HIPK2*, *SEMA4D*, *TREM1*, *VEGFA*, *HAVCR2*, *LST1*, *OLR1*, *GK*, *CD4*, *HCLS1*, *TLR1*, *CD74*, *PLXDC2*, *HLA-DMB*, *MAFB*, *TYROBP*, *RNASE6*, *NR4A2*, *ARHGAP18*, *PLEK*, *METRNL*, *DUSP4*, *HLA-DQA1*, *DNAJC15*, *LYZ*
	C8	TRAVAGLINI_LUNG_CILIATED_CELL	0.0002	0.0155	0.393	2.460	31	*DSP*, *RGS22*, *CKB*, *OAS2*, *IFI6*, *HES2*, *GBP3*, *NELL2*, *CD4*, *PPME1*, *CTSS*, *KLHL6*, *ARHGAP18*, *DNER*, *P4HA2*, *RARRES1*, *DUSP4*, *ACAP1*, *GPC3*, *PLAC8*, *RTP4*
	C8	TRAVAGLINI_LUNG_TREM2_DENDRITIC_CELL	0.0002	0.0155	0.293	2.334	57	*HLA-DOA*, *FCHO2*, *CSF1R*, *IFI6*, *TREM1*, *HAVCR2*, *CEBPA*, *LGALS9*, *OLR1*, *GK*, *CD4*, *FBP1*, *CD74*, *CTSS*, *PLXDC2*, *HLA-DMB*, *MAFB*, *EIF4EBP1*, *STAT1*, *TYROBP*, *RNASE6*, *ARHGAP18*, *CXCL10*, *RARRES1*, *GPNMB*, *LY86*, *HLA-DQA1*, *SLC7A7*, *LYZ*, *SLC40A1*
	C8	AIZARANI_LIVER_C25_KUPFFER_CELLS_4	0.0005	0.0265	0.375	2.350	31	*HLA-DOA*, *CSF1R*, *VEGFA*, *LGALS9*, *LST1*, *HCLS1*, *CTSS*, *HLA-DMB*, *MAFB*, *KLHL6*, *TYROBP*, *RNASE6*, *NR4A2*, *PLEK*, *CD55*, *METRNL*, *HLA-DQA1*, *LYZ*, *GZMB*, *SAMSN1*, *ITGB2*
	C8	TRAVAGLINI_LUNG_NATURAL_KILLER_CELL	0.0007	0.0276	0.448	2.357	22	*CD247*, *IL2RB*, *HAVCR2*, *AOAH*, *GPR65*, *PYHIN1*, *ITGAL*, *HSH2D*, *GNLY*, *CTSW*, *KLRD1*, *IFITM1*, *KLRB1*, *PLAC8*, *CCL5*, *GZMB*, *ITGB2*
	C8	TRAVAGLINI_LUNG_MACROPHAGE_CELL	0.0009	0.0315	0.440	2.264	21	*IFI6*, *TREM1*, *IL6*, *OLR1*, *GK*, *FBP1*, *CD74*, *CTSS*, *PLXDC2*, *TYROBP*, *GPNMB*, *HLA-DQA1*, *SLC7A7*, *LYZ*
	C8	TRAVAGLINI_LUNG_LIPOFIBROBLAST_CELL	0.0013	0.0356	0.507	2.236	14	*PITPNC1*, *VEGFA*, *PVT1*, *CD74*, *NR4A2*, *RARRES1*, *GPNMB*, *IFITM1*, *LONRF2*, *LDLR*, *HK2*
	C8	HAY_BONE_MARROW_NK_CELLS	0.0015	0.0356	0.300	2.220	46	*KIR2DL3*, *PITPNC1*, *CD247*, *IL2RB*, *HAVCR2*, *HCST*, *AOAH*, *TBC1D10C*, *TSPAN32*, *IL18RAP*, *FEZ1*, *GPR65*, *PYHIN1*, *ITGAL*, *CERCAM*, *EOMES*, *HLA-F*, *HSH2D*, *GNLY*, *DOK2*, *CTSW*, *KLRD1*, *IFITM1*
	C8	AIZARANI_LIVER_C2_KUPFFER_CELLS_1	0.0015	0.0356	0.361	2.148	28	*HLA-DOA*, *CSF1R*, *LGALS9*, *LST1*, *HCLS1*, *CTSS*, *PLXDC2*, *HLA-DMB*, *MAFB*, *KLHL6*, *TYROBP*, *RNASE6*, *CD37*, *PLEK*, *LY86*, *HLA-DQA1*, *LYZ*

## Data Availability

The original datasets are publicly available from NCBI as outlined in Section 2. The R codes will be shared upon request.

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
