# Peer review of "Unsupervised Analysis Reveals the Involvement of Key Immune Response Genes and the Matrisome in Resistance to BRAF and MEK Inhibitors in Melanoma"

_cancers, 2024, doi:10.3390/cancers16132313_

Round 1

Reviewer 1 Report (Previous Reviewer 2)

Comments and Suggestions for Authors

The manuscript has been improved by the addition of experimental validation. Please include statistical analysis for these new results.

Author Response

Thank you for the suggestion.  We have now included in the "Method" section of the statistical method for calculating p values for the qRT-PCR experiments. These p values are also included in the Figure 2, as well as the results text.  

This manuscript is a resubmission of an earlier submission. The following is a list of the peer review reports and author responses from that submission.

Round 1

Reviewer 1 Report

Comments and Suggestions for Authors

The manuscript, by Liu-Smith F., aims to report genes and pathways that can be targeted for further studies, with the ultimate goal of their eventual use in the clinic. The author pursued this aim by analyzing two sets of previously published tumor genomics data, comparing pre-treated melanoma and BRAFi and/or MEKi-resistant tumors. The study was conducted using statistical analytical measurements and bioinformatic tools (Gene Set Enrichment Analysis, Principal Component Analysis (PCA), and Differentially Expressed Gene (DEG). The author's findings revealed the heterogeneity of the tumors in response to BRAFi and BRAFi/MEKi treatment, as indicated by the inability to generate clusters. Using DEG analysis, the two data sets shared 8 genes and 2 sets of signature genes (matrisome and matrisome-associated signature). The author reported the relevance of these genes to tumor microenvironment and immune response, shedding light on their potential therapeutic value. Notably, PLXNC1, shared between data sets and upregulated in resistant tumors, emerged as a key focus of the study.

In conclusion, the author identified the described shared immune- and matrisome-related genes as potential therapeutic targets. Further validating the meticulousness and robustness of the study is recommended. The study's question of heterogeneity and finding a common therapeutic target is interesting. However, it would be more informative if accomplished using a larger data set and validated by experimental approaches.

Suggestions:

- Typos are found in some places. Here are two examples: Line 28: microenviroment; Line 63: Principle

- Figure 1: legend needs more information. It would be better if the author made it more informative. For example, the patient bar needs to be described; it might miss numbers; does the same color refer to tumors collected from the same patient?

- Figure 1: It would be better if the abbreviations were written in full within the legend or in an abbreviation list, particularly the response to treatment.

- Figure 1, It would be helpful to make it easier to read; for example, change the very close color code of some parameters (PR/SD), patients (missing numbers)

- Figure 1: Panel labeling (capitalized in the text and small caps in the figure)

- Figure 1C shows multiple data for some patients (e.g., pretreated patients 11, 11; progressed patients 11, 11); it needs to be described whether they are various clusters of the same tumor or if each is a different tumor cluster.

- Table 1 is difficult to follow. The formatting needs to be adjusted, particularly the column for the pathway name.

- Methods: Were each patient's tumors represented by progressed and pre-treated tumors? Or did the study include either pretreated or progressed tumors from some patients?

- Result: Which type of treatment was Patient 10 on? It should be mentioned in line 89, where the overlap between original and progressed tumors was stated. Does this overlap match the degree of response that the patient had?

- Results: Section “Analysis using the melanoma-specific gene set also returned no significant clustering.” These results were not linked to the presented data; is the data not shown, or can it be summarized in a table?

- Discussion: The inflammatory microenvironment reported in resistant tumors needs to be discussed; commonly, inflammatory events are inhibited in resistant tumors; however, they could be relatively active in correlation to provided treatment.  

- Discussion: The previously described role of PLXNC1 in melanoma should be discussed and cited. It is mostly defined as a tumor suppressor gene. The current study detected the upregulation of PLXNC1 in resistant tumors. Does it correlate with the response to treatment? Also, the upregulation of PXNC1 has been reported in other types of cancers, such as HCC, so including this in the discussion makes PXNC1 more valuable as a therapeutic target.

-Discussion: The study would be more interesting if it included data for tumors collected from patients treated with immunotherapies and/or others who received a combination of drug-targeted treatment and immunotherapies. The question here is: Has the heterogeneity decreased? Are there different sets of potential therapeutic targets?

Author Response

In conclusion, the author identified the described shared immune- and matrisome-related genes as potential therapeutic targets. Further validating the meticulousness and robustness of the study is recommended. The study's question of heterogeneity and finding a common therapeutic target is interesting. However, it would be more informative if accomplished using a larger data set and validated by experimental approaches.

Response: Thank you for your kind review. We agree that a larger dataset would be more informative and validation would be required in the future. This is a limitation for the current study and discussion is discussed in the “discussion” section (line 229-232)

Suggestions:

- Typos are found in some places. Here are two examples: Line 28: microenviroment; Line 63: Principle

Response: A grammar check was performed and these errors are now corrected. Additionally the MSigDb (line 129) was changed to MSigDB to be consistent with the rest of text.

- Figure 1: legend needs more information. It would be better if the author made it more informative. For example, the patient bar needs to be described; it might miss numbers; does the same color refer to tumors collected from the same patient?

Response: The patient number was cut off and now it is revealed. No, the tumor color scheme means pretreated or progressed. Descriptions have been added in the legend.

- Figure 1: It would be better if the abbreviations were written in full within the legend or in an abbreviation list, particularly the response to treatment.

Response: Abbreviations were spelled out.

- Figure 1, It would be helpful to make it easier to read; for example, change the very close color code of some parameters (PR/SD), patients (missing numbers)

Response: Corrected, PR is now changed to blue.

- Figure 1: Panel labeling (capitalized in the text and small caps in the figure)

Response: Changed to small caps in the text and in the legend, thanks.

- Figure 1C shows multiple data for some patients (e.g., pretreated patients 11, 11; progressed patients 11, 11); it needs to be described whether they are various clusters of the same tumor or if each is a different tumor cluster.

Response: Each is a different tumor from the same patients. It is now described in the legend.

- Table 1 is difficult to follow. The formatting needs to be adjusted, particularly the column for the pathway name.

Response: I believe the reviewer refers to Table 3 here. So Table 3 is adjusted in the format.

- Methods: Were each patient's tumors represented by progressed and pre-treated tumors? Or did the study include either pretreated or progressed tumors from some patients?

Response: Some patients have multiple tumors either before or after treatment. A detailed table is added in the supplemental materials (now Table S1), and also indicated in the text (line 79-80)

- Result: Which type of treatment was Patient 10 on? It should be mentioned in line 89, where the overlap between original and progressed tumors was stated. Does this overlap match the degree of response that the patient had?

Response:  Patient 10 was treated with Dabrafenib, and had a partial response. This info is now mentioned in results.

- Results: Section “Analysis using the melanoma-specific gene set also returned no significant clustering.” These results were not linked to the presented data; is the data not shown, or can it be summarized in a table?

Response: These results are not shown as they are identical to the ones that are shown, as stated in the text:” The results were identical to the first analysis, with no apparent clustering of pre-treated tumors and resistant tumors.”

- Discussion: The inflammatory microenvironment reported in resistant tumors needs to be discussed; commonly, inflammatory events are inhibited in resistant tumors; however, they could be relatively active in correlation to provided treatment.  

Response: A paragraph is added to address this issue in the discussion.

- Discussion: The previously described role of PLXNC1 in melanoma should be discussed and cited. It is mostly defined as a tumor suppressor gene. The current study detected the upregulation of PLXNC1 in resistant tumors. Does it correlate with the response to treatment? Also, the upregulation of PXNC1 has been reported in other types of cancers, such as HCC, so including this in the discussion makes PXNC1 more valuable as a therapeutic target.

Response: A paragraph is added to discuss this issue.

-Discussion: The study would be more interesting if it included data for tumors collected from patients treated with immunotherapies and/or others who received a combination of drug-targeted treatment and immunotherapies. The question here is: Has the heterogeneity decreased? Are there different sets of potential therapeutic targets?

Response: This is an interesting question. Based on Figure 1a, judging from the distribution, heterogeneity seems to be increased in the progressed tumors (judging by eyeball but actual statistic is needed to evaluate the degree of heterogeneity). In the future study we will look into these questions.

Reviewer 2 Report

Comments and Suggestions for Authors

An article by Liu-Smith demonstrates the issue of drug resistance in melanoma patients after the use of BRAFi/MEKi therapy. As the tumor heterogeneity and heterogeneity in treatment response made it difficult to find consensus genes and pathways in the resistance to therapy, this study used an objective method to analyze published gene expression data from pre-treated tumors and drug-resistant tumors, and identified possible targets and markers for resistant tumors, which is centered at PLXNC1 which promotes a pro-inflammatory tumor microenvironment.

Specific comments:

1. Gene names should be consistently written in italics.

2. The major drawback of this study is that it is exclusively bioinformatic analysis, and it is not supported by any experimental validation. The Author should consider to assess e.g., PLXNC1 level at mrna and/or transcript level in drug-resistant cell lines.

3. The matrisome should be more extensively defined and described in the introduction.

Author Response

Thank you for you kind review, the following points have been addressed:

  1. Gene names should be consistently written in italics.

Response: all genes are italicized now. Thank you.

  1. The major drawback of this study is that it is exclusively bioinformatic analysis, and it is not supported by any experimental validation. The Author should consider to assess e.g., PLXNC1 level at mrna and/or transcript level in drug-resistant cell lines.

Response: This is a good point. The PI will follow this study and investigate the role of PLXNC1 in tumors and melanoma cells but that will be our future direction.

  1. The matrisome should be more extensively defined and described in the introduction.

Response: A paragraph is added in the introduction to describe matrisome. Thank you.

Round 2

Reviewer 1 Report

Comments and Suggestions for Authors

The author generated a better-revised version of the manuscript. Future follow-up to validate the study would be beneficial.

Reviewer 2 Report

Comments and Suggestions for Authors

My major comment on the lack of experimental validation has not been addressed.
